# Molecular Detection of Various Non-Seasonal, Zoonotic Influenza Viruses Using BioFire FilmArray and GenXpert Diagnostic Platforms

**DOI:** 10.3390/v17070970

**Published:** 2025-07-10

**Authors:** Charlene Ranadheera, Taeyo Chestley, Orlando Perez, Breanna Meek, Laura Hart, Morgan Johnson, Yohannes Berhane, Nathalie Bastien

**Affiliations:** 1National Microbiology Laboratory Branch, Public Health Agency of Canada, Winnipeg, MB R3E 3R2, Canada; charlene.ranadheera@phac-aspc.gc.ca (C.R.); taeyo.chestley@phac-aspc.gc.ca (T.C.); orlando.perez@phac-aspc.gc.ca (O.P.); breannaimeek@gmail.com (B.M.); laura.hart@phac-aspc.gc.ca (L.H.); mkat37@gmail.com (M.J.); 2National Centre for Foreign and Animal Disease, Canadian Food Inspection Agency, Winnipeg, MB R3E 3M4, Canada; yohannes.berhane@inspection.gc.ca; 3Department of Veterinary Pathology, Western College of Veterinary Medicine, University of Saskatchewan, Saskatoon, SK S7N 5B4, Canada; 4Department of Pathobiology, University of Guelph, Guelph, ON NIG 2W1, Canada; 5Department of Animal Science, University of Manitoba, Winnipeg, MB R3T 2N2, Canada

**Keywords:** avian influenza, zoonotic influenza, pandemic preparedness, point of care testing

## Abstract

Since 2020, the Gs/Gd H5N1 influenza virus (clade 2.3.4.4b) has established itself within wild bird populations across Asia, Europe, and the Americas, causing outbreaks in wild mammals, commercial poultry, and dairy farms. The impacts on the bird populations and the agricultural industry has been significant, requiring a One Health approach to enhanced surveillance in both humans and animals. To support pandemic preparedness efforts, we evaluated the Cepheid Xpert Xpress CoV-2/Flu/RSV plus kit and the BioFire Respiratory 2.1 Panel for their ability to detect the presence of non-seasonal, zoonotic influenza A viruses, including circulating H5N1 viruses from clade 2.3.4.4b. Both assays effectively detected the presence of influenza virus in clinically-contrived nasal swab and saliva specimens at low concentrations. The results generated using the Cepheid Xpert Xpress CoV-2/Flu/RSV plus kit and the BioFire Respiratory 2.1 Panel, in conjunction with clinical and epidemiological findings provide valuable diagnostic findings that can strengthen pandemic preparedness and surveillance initiatives.

## 1. Introduction

Since the COVID-19 pandemic, investment into pandemic preparedness has been a central focus of public health initiatives. Historically, influenza viruses have drawn significant attention of public health authorities due to their pandemic threat. In 1918, the H1N1 Spanish Flu global pandemic is estimated to have caused the deaths of approximately 100 million people worldwide [1,2,3]. The 1957 H2N2 Asian and 1968 H3N2 Hong Kong Flus were each responsible for approximately 1–4 million deaths [4]. In 1977, the H1N1 Russian Flu led to approximately 700,000 deaths [5] and the most recent 2009 H1N1 Swine Flu pandemic was responsible for upwards of 575,000 deaths [6]. To support mitigation and preparedness efforts, influenza global surveillance efforts have been established to monitor circulating strains in both humans and animals [7,8].

In 2020, the Gs/Gd lineage (A/goose/Guangdong/1/1996) of H5 clade 2.3.4.4b viruses reemerged and subsequently spread throughout Asia, Europe, and the Americas [9,10]. Since then, it has established itself within wild bird populations and led to outbreaks in commercial poultry and dairy farms [11,12,13,14]. There has been a remarkably high number of H5 clade 2.3.4.4b human infections in North America with the majority of these cases presenting with mild symptoms; however, some infections have resulted in severe respiratory complications and even death [14,15,16,17,18,19]. The ability of the virus to transmit from birds to various mammalian species has caused concern that this virus could mutate and adapt, and eventually lead to sustained transmission within humans.

In Canada, seasonal influenza molecular diagnostic testing is decentralized across laboratories nationwide. Testing platforms can range from high-throughput laboratory-developed molecular tests to low-throughput commercial point of care (POC) testing. Commercial POC or near-POC platforms have been distributed widely across Canada to better support northern, remote and isolated communities. Many commercial POC or near POC testing platforms effectively detect influenza A, but only a limited number of manufacturers offer HA and/or NA subtyping capabilities, and are often limited to the detection of seasonal influenza H1N1 or H3N2 subtypes. The identification of non-seasonal influenza strains of zoonotic origin remains a key priority for pandemic preparedness and often requires further characterization to identify subtype, HA clade designation, and viral genotype. Therefore, considerations should be made to detect and subtype the virus from all cases of influenza to ensure appropriate patient care and containment.

The Cepheid Xpert^®^ Xpress CoV-2/Flu/RSV plus kit uses PCR-based technology to detect the presence of SARS-CoV-2, Influenza A and B and/or respiratory syncytial virus (RSV) within a specimen. The BioFire^®^ Respiratory 2.1 Panel employs nested PCR technology to assess a specimen for the presence of 22 different targets from a variety of pathogens causing respiratory illness. More specifically, this panel distinguishes between influenza A and B viruses, and within the former can subtype seasonal H1 and H3 viruses. However, these platforms have not been fully assessed for the detection of non-seasonal, zoonotic influenza, due to the lack of clinical specimens available.

To support pandemic preparedness and ensure comprehensive human diagnosis of influenza virus infection, this study evaluates the capability of the Cepheid Xpert^®^ Xpress CoV-2/Flu/RSV plus kit and the BioFire^®^ Respiratory 2.1 Panel to effectively detect the presence of non-seasonal, zoonotic influenza A viruses in contrived clinical specimens and for the first time, currently circulating H5N1 strains have been assessed.

## 2. Materials and Methods

### 2.1. Cells and Viruses

MDCK-CCL-34 cells (American Type Culture Collection, Manassas, VA, USA) were used to propagate various influenza virus strains. In short, MDCK cells were grown in minimum essential medium (MEM) supplemented with 10% fetal bovine serum, and 1% L glutamine (ThermoFisher Scientific, Mississauga, ON, Canada). Cells were seeded 24 h prior to infection so that they were 70% confluent at the time of infection. Cells were washed with phosphate buffered saline and replaced with MEM supplemented with 0.1% bovine serum albumin, 1% L glutamine and 1 µg/mL TPCK trypsin (ThermoFisher Scientific, Mississauga, ON, Canada). Working in a containment level three laboratory, cells were infected with influenza at a multiplicity of infection of 0.001 and supernatants were harvested when ≥80% cytopathic effect was observed. Viruses were inactivated with 3–5 megarads of gamma irradiation, tested to confirm infectious material was not present in the specimens and subsequently transferred into a containment level two laboratory. Strain-specific synthetic matrix gene (M-gene) RNA was used to quantify the copies of genomic RNA in each contrived specimen using a real-time reverse transcriptase PCR assay previously published [20]. The forward primer (AF_deg_) was modified to include two degenerate “Y” (pyrimidine) nucleotides to account for strain-specific variability (Table 1).

### 2.2. Contrived-Clinical Specimens, Result Interpretation and Limit of Detection

A number of zoonotic influenza viruses were selected to encompass a wide breadth of strains with pandemic potential (Table 2). Inactivated virus was serially diluted and added to either a pooled human nasal swab specimen background matrix [21] or a pooled human saliva background matrix (Medix Biochemica, Montana, USA) to mimic a clinical specimen. According to the respective manufacturer’s directions, 300 µL of the contrived-clinical specimen was added to either the Xpert^®^ Xpress CoV-2/Flu/RSV plus kit (Cepheid, Sunnyvale, CA, USA) or the BioFire^®^ Respiratory 2.1 Panel (bioMérieux, Montreal, QC, Canada). The Xpert^®^ Xpress CoV-2/Flu/RSV plus kit generates either a “positive” or “negative” result for the detection of influenza A (Flu A) virus. A positive result indicates that at least one out of two Flu A targets passed the assay requirements [22]. The BioFire^®^ Respiratory 2.1 Panel provides a “positive”, “negative”, or “equivocal” result for the detection of influenza A viruses. According to the manufacturer’s results interpretation, a positive result occurs when at least one out of the two Flu A pan assays are positive in conjunction with an H1 or H3 subtyping assay, or both Flu A pan assays are positive with no subtyping assay. An equivocal result is when one out of the two Flu A pan assays are positive or when a subtype assay is positive without a positive Flu A pan assay [23]. Since the H5, H7, H9 and H10 assays are not expected to interact with the subtyping assays, and using a similar interpretation strategy as the Xpert^®^ Xpress CoV-2/Flu/RSV plus kit, a positive result for the limit of detection will also include any equivocal results where one out of the two Flu A pan assays is positive. Each sample was tested with six replicates and the lowest dilution yielding 100% positivity was determined to be the limit of detection.

## 3. Results

### 3.1. Limit of Detection of Non-Seasonal, Zoonotic Influenza Viruses in Nasal Swab Specimens

Contrived-clinical nasal swab samples were tested using the Cepheid Xpert Xpress CoV-2/Flu/RSV plus kit (Cepheid, Sunnyvale, CA, USA) and the BioFire Respiratory 2.1 Panel (bioMérieux, Montreal, QC, Canada). The Cepheid Xpert Xpress CoV-2/Flu/RSV plus kit successfully detected all tested non-seasonal, zoonotic strains and correctly distinguished between influenza A, influenza B, and RSV. However, a false positive result for SARS-CoV-2 was observed in the specimen containing the H3N2 seasonal strain, although the test did correctly identify the presence of an influenza A virus (Table 3). The limit of detection (LOD) for the Cepheid Xpert Xpress CoV-2/Flu/RSV plus kit ranged from 210 to 8000 copies/mL depending on the strain being tested (Table 3). The BioFire Respiratory 2.1 panel also correctly distinguished between the presence of influenza A strains and the other respiratory pathogens on this panel. In addition, it was able to correctly subtype all of the H3 viruses tested and was able to subtype one out of two H1 viruses (Table 3). The assay was not able to subtype the H1N2 variant (A/Manitoba/01/2021) that was responsible for a single zoonotic transmission event from swine into a human. The LOD for the BioFire Respiratory 2.1 ranged from 240 to 80,000 copies/mL between strain types (Table 3).

### 3.2. Limit of Detection of Non-Seasonal, Zoonotic Influenza Viruses in Saliva Specimens

To evaluate the ability of these platforms to process alternative specimen types, a second set of contrived-clinical samples was prepared by combining known amounts of inactivated virus into a saliva background, which is a possible specimen matrix for the detection of various respiratory viruses [24,25,26]. Both platforms performed similarly to the specimens in a nasal background. The Cepheid Xpert Xpress CoV/Flu/RSV/plus assay was able to accurately detect the presence of influenza A in all samples tested (Table 4). The strain-dependent LOD ranged from 27 to 8000 copies/mL (Table 4). Likewise, the BioFire Respiratory 2.1 panel distinguished influenza A from the other respiratory pathogen targets, while also providing subtyping for the H3 viruses and one of the two H1 viruses (Table 4). The LOD ranged from 240 to 80,000 copies/mL (Table 4).

Both assays were effectively able to detect the zoonotic influenza strains with a sensitivity comparable to the seasonal H1N1 strain used in this study, with an observed variation of approximately 1log_10_. In a nasal specimen, the Xpert Xpress CoV-2/Flu/RSV plus assay had increased sensitivity over the BioFire Respiratory 2.1 panel; typically, with a 1log_10_ improvement in LOD across all strains except for the H5N1-duck and H10N7-seal viruses where they performed comparably (Table 3). However, in saliva specimens, the BioFire Respiratory 2.1 had improved LODs for several viruses, the H3N2-canine, all H5N1 and H10N7 viruses, aligning with results obtained using the Xpert Xpress CoV-2/Flu/RSV plus assay (Table 4). The BioFire Respiratory 2.1 panel demonstrated greater specificity than the Cepheid Xpert Xpress CoV-2/Flu/RSV plus assay, through its additional subtyping capacity and sample concordance between pathogens. While the subtyping assays may not be able to detect all zoonotic-H1 and -H3 viruses, the H1 and H3 assays did not cross react with the other subtypes used in this study.

## 4. Discussion

The pandemic potential of the recently established influenza H5N1 (clade 2.3.4.4b) strain in wild birds has led to heightened vigilance among public health authorities, demonstrating the need for greater diagnostic and enhanced surveillance capacities in both humans and animals to rapidly diagnose infections, distinguish between seasonal and zoonotic influenza infections, and track strains exhibiting pandemic potential characteristics.

The use of all-in-one testing platforms has become increasingly widespread to support patient diagnostics. While not the most appropriate platform for high-throughput testing, these low-throughput diagnostic systems have several advantages that include reducing patient result turnaround times by providing near POC testing, and the ability to distinguish between a series of pathogens which have similar clinical presentations in a single run; the user-friendly nature of these platforms minimizes the technical expertise and the hands-on time required by staff, allowing better capacity building across multiple disciplines and geographic regions, particularly when local testing is not normally available. For example, in Canada, these platforms are routinely used in hospital settings to support rapid diagnostic screening of patients experiencing severe acute respiratory illness, and in northern, remote and isolated communities, such as nursing stations, community hospitals, and provincial/territorial public health laboratories, to support more timely delivery of test results, alleviating the need for long-distance transportation to major centers and difficulties associated in maintaining cold chain and the subsequent integrity of the specimens. While these platforms have an important role in diagnostic testing and filling necessary gaps, these diagnostic assays are costly and may pose a financial strain for testing institutions if a funding mechanism is not in place.

The Cepheid GeneXpert and the BioFire FilmArray are two platforms that have been broadly adopted and have a diverse selection of pathogen test panels. Currently, the Cepheid Xpert Xpress CoV-2/Flu/RSV plus and the BioFire Respiratory 2.1 platforms are widely used for the detection of seasonal influenza strains but their ability to detect non-seasonal, zoonotic influenza strains has not been fully evaluated. Furthermore, the lack of clinical specimens available has impacted the ability to evaluate the suitability of these platforms for the detection of non-seasonal, zoonotic influenza strains in a more wholesome manner. Therefore, this study assessed the potential of these two near POC platforms to detect non-seasonal, zoonotic strains of influenza, including currently circulating influenza H5N1 (clade 2.3.4.4b) strains, in contrived nasal swab and saliva specimens.

The Cepheid GenXpert and the BioFire FilmArray technologies effectively processed both nasal swab and saliva samples, yielding comparable LODs. While both platforms demonstrated successful performance with these specimen types, selecting the appropriate sample type should always be guided by the specific shedding patterns of the pathogen being tested. Typically, a nasopharyngeal swab is the ideal sample type for molecular diagnostics of influenza virus; however, saliva samples have been used to detect the presence of various respiratory viruses, such as seasonal influenza, RSV and SARS-CoV-2, in POC or near POC settings [24,25,26,27,28,29,30].

Limited information is available in the literature evaluating the sensitivity and specificity of these platforms for detecting non-seasonal, zoonotic strains with much of the existing data provided by the manufacturer’s documentation. The manufacturer’s LOD determination for the BioFire Respiratory 2.1 panel was on average 330 copies/mL (0.5 TCID50/mL) for the influenza A assays [23], which was approximately three-fold lower than the LOD observed for seasonal influenza in this study. When the manufacturer conducted specificity testing of various other non-seasonal, zoonotic strains, including H1N2, H3N2v, H5N1 and H7N9 subtypes similarly used in this study, at three-fold the LOD value they were able to detect an influenza A positive result (without subtype), demonstrating the specificity of the assay [23]. Chan et al., conducted a study investigating avian and swine derived influenza viruses and their ability to be detected using the BioFire Respiratory 2.0 panel, which runs the same influenza assays as its successor. The authors obtained LODs for similar subtypes assessed here, H3N2v, H5N1, H7N9, H9N2 and H10N7 ranging from 3.6 to 60 TCID50/mL [31] and are 2.4 to 40-fold higher than the three-fold LOD concentration tested by the manufacturer’s specificity assessment. The present study produced LOD values ranging from 2.4 × 10^2^ to 6.2 × 10^4^ copies/mL for the non-seasonal, zoonotic strains, which range from 4.1-fold lower to 62-fold higher than the 3 three-fold LOD tested during the manufacturer’s specificity assessment.

The manufacturer of Cepheid Xpert Xpress CoV-2/Flu/RSV plus assay determined the LOD of their influenza A assay to be 0.007 TCID50/mL or 0.44 FFU/mL [22]. The manufacturer used viral RNA purified from avian influenza viruses in their assessment, which included similar influenza subtypes presented in this study, H5N1, H7N9, and H9N2. When tested at >1 pg/µL, all purified RNA isolates were detected by the assay [22]. Unfortunately, these units of quantification cannot be used to directly compare the LODs presented in this study. However, other studies assessed the LOD values of this assay using seasonal influenza A strains and determined that it ranged between approximately 50 and 100 copies/mL [32,33], which is in line with the seasonal results obtained in this study. The data presented here, for the first time, determine that the LOD values for the Cepheid Xpert Xpress CoV-2/Flu/RSV plus assay were between 27 and 6200 copies/mL for the non-seasonal, zoonotic strains, and at best are in line and at worst have a ten-fold higher LOD compared to what has been observed for seasonal strains. Collectively, this study provides important analytical data for both platforms and represents the first direct comparison of their capabilities to effectively detect non-seasonal, zoonotic strains.

Each platform presented with its own strengths and limitations. The Cepheid Xpert Xpress CoV-2/Flu/RSV plus assay had 10-fold improved sensitivity over the BioFire Respiratory 2.1 panel. However, it is important to note that the Cepheid Xpert Xpress CoV-2/Flu/RSV assay utilizes a 45-cycle amplification program versus the 40-cycle program in the BioFire Respiratory 2.1 assay. While the increased cycles may enhance sensitivity, it has been associated with a loss of specificity as evident by a discordant SARS-CoV-2 detection. Cross-reactivity between targets was not observed using the BioFire Respiratory 2.1, demonstrating increased specificity over the Cepheid Xpert Xpress CoV-2/Flu/RSV plus assay. In addition, the BioFire Respiratory 2.1 panel also had capabilities to subtype H3 and H1 viruses to varying degrees. The lack of subtyping of an influenza isolate by the BioFire Respiratory 2.1 panel could provide indicators that a non-seasonal, zoonotic strain is present and could be a mechanism used to triage this sample for further characterization. However, it should be noted that the detection of H1 or H3 subtypes would not rule out the possibility of a non-seasonal, zoonotic strain, since the targets were able to cross react with non-seasonal, zoonotic strains such as the H3N2-canine and H3N2v-human. Likewise, the detection of Flu A using the Cepheid Xpert Xpress CoV-2/Flu/RSV plus assay does not provide any indicators that the influenza detected is seasonal, or non-seasonal, zoonotic. Therefore, in these cases both clinical and epidemiological factors should be considered and factored in when determining if further characterization is necessary. Another consideration should be made for equivocal results generated by the BioFire Respiratory 2.1 panel; it is possible that as viruses drift and/or shift, only one of the two Flu A targets may be detected effectively. As such, an equivocal result could indicate the presence of a zoonotic virus or a strain with pandemic potential, providing an opportunity to triage specimens for further characterization. In general, both platforms are able to detect various non-seasonal, zoonotic strains, and in particular it is able to detect the new H5N1 H5 clade 2.3.4.4b virus, supporting their value in diagnostic and surveillance efforts and enhancing patient care.

## 5. Conclusions

This study provides the first direct analytical comparison of the Cepheid Xpert Xpress CoV-2/Flu/RSV Plus and BioFire Respiratory 2.1 platforms for their ability to detect non-seasonal, zoonotic influenza A strains, including the currently circulating H5N1 (clade 2.3.4.4b) viruses. Both platforms demonstrated the capacity to accurately detect a broad range of zoonotic influenza viruses in clinically relevant nasal swab and saliva specimens, with performance metrics that align with or complement existing manufacturer-reported values for seasonal strains. Furthermore, each platform offers unique advantages: the Cepheid Xpert system showed greater analytical sensitivity, while the BioFire Respiratory Panel provided added specificity through its subtyping capabilities. The lack of subtyping or presence of equivocal results may serve as important diagnostic indicators for potential zoonotic or emerging pandemic strains and could be leveraged as triggers for additional characterization. Having decentralized influenza diagnostic capacities is particularly important when circulating strains have been classified as having pandemic potential risk, non-seasonal, zoonotic or causing uncharacteristic clinical presentation.

## Figures and Tables

**Table 1 viruses-17-00970-t001:** M-gene oligo sequences for respective subtype quantification.

Assay Number	Subtype (Target)	Oligo Sequence
1	H1N1 (M)	AGACAAGACCAAUCUUGUCACCUCUGACUAAGGGAAUUUUAGGAUUUGUGUUCACGCUCACCGUGCCCAGUGAGCGAGGACUGCAGCGUAGACGCUUUAUCCAAAAUGCCCUAAAUG
2	H1N2v (M)	AGACAAGACCAAUCUUGUCACCUCUGACUAAGGGAAUUUUAGGAUUUGUGUUCACGCUCACCGUGCCCAGUGAGCGAGGACUGCAGCGUAGACGCUUUAUCCAAAAUGCCCUAAAUG
3	H3N2 (M)	AGACAAGACCAAUUCUGUCACCUUUGACUAAGGGGAUUUUAGGGUUUGUUUUCACGCUCACCGUGCCCAGUGAGCGAGGACUGCAGCGUAGACGCUUUGUCCAAAAUGCCCUCAAUG
4	H3N2v (M)	AGACAAGACCAAUCUUGUCACCAUUGACUAAGGGUAUUUUAGGAUUUGUGUUCACGCUCACCGUGCCCAGUGAGCGAGGACUGCAGCGUAGACGCUUUGUCCAAAAUGCCCUAAAUG
5	H5N1 (M)	AGACAAGACCAAUCCUGUCACCUCUGACUAAAGGAAUCUUGGGAUUUGUAUUCACGCUCACCGUGCCCAGUGAGCGAGGACUGCAGCGUAGACGUUUUGUUCAGAAUGCCCUAAAUG
6	H7N9 (M)	AGACAAGACCAAUCCUGUCACCUCUGACUAAGGGGAUUUUAGGGUUUGUGUUCACGCUCACCGUGCCCAGUGAGCGAGGACUGCAGCGUAGACGGUUUGUCCAAAACGCCCUAAAUG
7	H9N2/H10N7 (M)	AGACAAGACCAAUCCUGUCACCUCUGACUAAGGGGAUUUUAGGAUUUGUGUUCACGCUCACCGUGCCCAGUGAGCGAGGACUGCAGCGUAGACGCUUUGUCCAAAAUGCCCUUAAUG
		**Modified Primer Sequence**
AF_deg_	5′ Primer	GACCRATCYTGTCACCTYTGAC

**Table 2 viruses-17-00970-t002:** Influenza A Strain identification and characteristics.

Name	Subtype	Strain	Seasonal Strain	Species of Virus Origin	Species of Isolation	Quantification Assay Number
H1N1-seasonal	H1N1	A/turkey/ON/FAV8-12/2020	Yes	Human	Turkey	1
H1N2v-human	H1N2v	A/Manitoba/01/2021	No	Swine	Human	2
H3N2-seasonal	H3N2	A/Darwin/6/2021	Yes	Human	Human	3
H3N2-canine	H3N2	A/canine/ON/N-69-1/2018	No	Canine	Canine	3
H3N2v-human	H3N2v	A/Manitoba/03/2021	No	Swine	Human	4
H5N1-human	H5N1	A/Alberta/1/2014	No	Avian	Human	5
H5N1-AGWT	H5N1	A/AmericanGreenWingedTeal/NovaScotia/ FAV133-3/2022	No	Avian	American Green Winged Teal	5
H5N1-duck	H5N1	A/Duck/Quebec/FAV1347-10/2022	No	Avian	Wigeon	5
H5N1-human-TX	H5N1	A/Texas/37/2024	No	Avian/Bovine	Human	5
H5N1-human-BC	H5N1	A/BritishColumbia/PHL-2032/2024	No	Avian	Human	5
H7N9-human	H7N9	A/Anhui/01/13	No	Avian	Human	6
H9N2-human	H9N2	A/chicken/Ghana/Slycat-13/2018	No	Avian	Human	7
H10N7-seal	H10N7	A/harbour seal/BC/OTH52-4/2021	No	Seal	Seal	7

**Table 3 viruses-17-00970-t003:** Limit of detection of contrived nasal samples of various influenza strains using the Cepheid Xpert Xpress CoV-2/Flu/RSV plus kit and the BioFire Respiratory 2.1 Panel.

Name	Cepheid Xpert^®^ Xpress CoV-2/Flu/RSV Plus		BioFire^®^ Respiratory 2.1 Panel	
	Positive Result	Limit of Detection (M copies/mL)	Positive Result and Subtype	Limit of Detection (M Copies/mL)
H1N1-seasonal	Flu A	2.1 × 10^2^	Flu A/H1	2.1 × 10^3^
H1N2v-human	Flu A	5.0 × 10^2^	Flu A	5.0 × 10^3^
H3N2-seasonal	Flu A * (SARS-CoV-2)	8.0 × 10^3^	Flu A/H3	8.0 × 10^4^
H3N2-canine	Flu A	9.0 × 10^2^	Flu A/H3	9.0 × 10^3^
H3N2v-human	Flu A	1.5 × 10^3^	Flu A/H3	1.5 × 10^4^
H5N1-human-AB	Flu A	4.7 × 10^3^	Flu A	4.7 × 10^4^
H5N1-AGWT	Flu A	2.4 × 10^3^	Flu A	2.4 × 10^4^
H5N1-duck	Flu A	2.4 × 10^2^	Flu A	2.4 × 10^2^
H5N1-human-TX	Flu A	2.4 × 10^3^	Flu A	2.4 × 10^4^
H5N1-human-BC	Flu A	8.3 × 10^2^	Flu A	8.3 × 10^3^
H7N9-human	Flu A	6.2 × 10^3^	Flu A	6.2 × 10^4^
H9N2-human	Flu A	2.7 × 10^2^	Flu A	2.7 × 10^3^
H10N7-seal	Flu A	2.4 × 10^3^	Flu A	2.4 × 10^3^

* Discordant result observed.

**Table 4 viruses-17-00970-t004:** Limit of detection of contrived saliva samples of various influenza strains using the Cepheid Xpert Xpress CoV-2/Flu/RSV plus kit and the BioFire Respiratory 2.1 Panel.

Name	Cepheid Xpert^®^ Xpress CoV-2/Flu/RSV Plus		BioFire^®^ Respiratory 2.1 Panel	
	Positive Result	Limit of Detection (M Copies/mL)	Positive Result and Subtype	Limit of Detection (M Copies/mL)
H1N1-seasonal	Flu A	2.1 × 10^2^	Flu A/H1	2.1 × 10^3^
H1N2v-human	Flu A	5.0 × 10^2^	Flu A	5.0 × 10^3^
H3N2-seasonal	Flu A	8.0 × 10^3^	Flu A/H3	8.0 × 10^4^
H3N2-canine	Flu A	9.0 × 10^2^	Flu A/H3	9.0 × 10^2^
H3N2v-human	Flu A	1.5 × 10^3^	Flu A/H3	1.5 × 10^4^
H5N1-human	Flu A	4.7 × 10^3^	Flu A	4.7 × 10^3^
H5N1-AGWT	Flu A	2.4 × 10^3^	Flu A	2.4 × 10^3^
H5N1-Duck	Flu A	2.4 × 10^2^	Flu A	2.4 × 10^2^
H5N1-human-TX	Flu A	2.4 × 10^3^	Flu A	2.4 × 10^4^
H5N1-human-BC	Flu A	8.3 × 10^2^	Flu A	8.3 × 10^3^
H7N9-human	Flu A	6.2 × 10^3^	Flu A	6.2 × 10^4^
H9N2-human	Flu A	2.7 × 10^1^	Flu A	2.7 × 10^3^
H10N7-seal	Flu A	2.4 × 10^3^	Flu A	2.4 × 10^3^

## Data Availability

Original data can be requested by contacting the corresponding author.

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
