# Peer review of "Molecular Detection of Various Non-Seasonal, Zoonotic Influenza Viruses Using BioFire FilmArray and GenXpert Diagnostic Platforms"

_viruses, 2025, doi:10.3390/v17070970_

Round 1
Reviewer 1 Report
Comments and Suggestions for Authors
This manuscript presents the first description of detecting non-seasonal, zoonotic influenza A viruses—including circulating H5N1 clade 2.3.4.4b strains—using commercially available PCR-based kits. The authors address an important gap by directly comparing two widely used commercial PCR-based diagnostic platforms (i.e. the Cepheid Xpert Xpress CoV-2/Flu/RSV plus and BioFire Respiratory 2.1) for their ability to detect these non-seasonal influenza strains.
The study is well designed and executed, effectively comparing detection performance on contrived pooled human nasal swab and saliva specimens. The Cepheid platform shows generally superior sensitivity, with approximately one-log lower limits of detection across many strains. However, the BioFire platform offers clear advantages in specificity, including subtyping capabilities for H1 and H3, which can aid in differentiating seasonal from non-seasonal strains and support epidemiological investigations. Therefore, this trade-off between sensitivity and specificity appears as a central consideration in this comparison, with significant implications for public health use.
While fully acknowledging the challenges of obtaining and handling clinical specimens from patients infected with non-seasonal influenza strains, the absence of testing such samples in this context makes it difficult to assess the real-world impact of the difference in limit of detection between the two platforms. Although the BioFire has a higher limit of detection - approximately one log less sensitive than the Cepheid - it remains unclear whether this difference has meaningful clinical implications. In real-world settings, typical viral loads in patient samples may exceed both platforms’ detection thresholds, rendering the additional sensitivity redundant for most cases. Conversely, if patients could actually display low viral load (early or late in infection, or with mild disease), the enhanced sensitivity of the Cepheid assay could be a significant advantage. Unfortunately, the absence of data on clinical specimens allowing to better estimate the relevant spectrum of viral loads limits our understanding of the practical advantage of the lower detection limit achieved by Cepheid. This point should be addressed in the Discussion section.
This work makes a valuable contribution by providing the first direct analytical comparison of these platforms for detecting potentially emerging zoonotic influenza strains. Wider deployment of such platforms could facilitate earlier detection of zoonotic transmission events, enable prompt containment measures, and better inform pandemic preparedness planning. Moreover, while such manuscript focusing on analytical performance evaluation may be regarded as less central in academic publishing priorities, it is important to acknowledge their role for ensuring that diagnostic platforms meet public health needs, guiding evidence-based deployment decisions, and ultimately supporting timely and effective outbreak responses.
Author Response
Comments 1: The absence of data on clinical specimens allowing to better estimate the relevant spectrum of viral loads limits our understanding of the practical advantage of the lower detection limit achieved by Cepheid. This point should be addressed in the Discussion section.
Response 1: Thank you for your comment. This is quite helpful and we feel will improve both the introduction and discussion by more clearly stating the roadblocks associated with validating these systems. The following changes have been made to both the introduction and discussion (changes in the manuscript have been tracked in red):
Lines 68-74: However, these platforms have not been fully assessed for the detection of non-seasonal, zoonotic influenza, due to the lack of clinical specimens available. To support pandemic preparedness and ensure comprehensive human diagnosis of influenza virus infection, this study evaluates the capability of the Cepheid Xpert® Xpress CoV-2/Flu/RSV plus kit and the BioFire® Respiratory 2.1 Panel to effectively detect the presence of non-seasonal, zoonotic influenza A viruses in contrived clinical specimens and for the first time, currently circulating H5N1 strains have been assessed.
Line 193-196: Furthermore, the lack of clinical specimens available has impacted the ability to evaluate the suitability of these platforms for the detection of non-seasonal, zoonotic influenza strains in a more wholesome manner.
Reviewer 2 Report
Comments and Suggestions for Authors
Dear authors,
The article to be published, Molecular detection of various non-seasonal, zoonotic influenza viruses using BioFire FilmArray and GenXpert Diagnostic 3 Platforms, is a current topic of great concern for public health authorities, considering that influenza is a widespread virus in wildlife. Therefore, it would be important to reflect in your work on whether these platforms would be financially accessible to public health services, since official notification is always provided by public agencies. Furthermore, the detection described in the Materials and Methods section refers to an inactivated and subsequently diluted virus. My question is whether, given that the virus is inactivated, it would not accurately reflect what happens in natural infections, considering that there is no way to determine the infectious load of the sample being tested. In any case, it is a well-prepared work with conclusions of interest to the scientific community. Congrats
Author Response
Comments 1: It would be important to reflect in your work on whether these platforms would be financially accessible to public health services, since official notification is always provided by public agencies
Response 1: Thank you for pointing this out. We agree with this comment and have modified the text accordingly. Changes have been tracked within the manuscript in red.
Line 180-188: For example in Canada, these platforms are routinely used in hospital settings to support rapid diagnostic screening of patients experiencing severe acute respiratory illness, and in northern, remote and isolates communities, such as nursing stations, community hospitals, and provincial/territorial public health laboratories, to support more timely delivery of test results alleviating the need for long-distance transportation to major centers and difficulties associated in maintaining cold chain and the subsequent integrity of the specimens. While these platforms have an important role in diagnostic testing and filling necessary gaps, these diagnostic assays are costly and may pose a financial strain for testing institutions if a funding mechanism is not in place.
Comments 2: The detection described in the Materials and Methods section refers to an inactivated and subsequently diluted virus. My question is whether, given that the virus is inactivated, it would not accurately reflect what happens in natural infections, considering that there is no way to determine the infectious load of the sample being tested.
Response 2: Thank you for your comment and we apologize if we have misunderstood the question being asked. Since we are employing a nucleic acid amplification test (NAAT) the infectivity of the virus is less relevant. Therefore, we use M genome copies/ml, indirectly inferring viral load, rather than directly tittering the specimen to determine IFU/ml. In our hands, the inactivation process does not affect the detection sensitivity of viral RNA using a NAAT. We have had difficulty reliability isolating virus from primary specimens with high Ct values (Cts >20). There is also the added benefit of inactivating the virus prior to extraction so that this testing can be done in biosafety level 2 rather than biosafety level 3.
Reviewer 3 Report
Comments and Suggestions for Authors
This study provides valuable data on the performance of the BIOFIRE® Respiratory 2.1 Panel and Xpert® Xpress CoV-2/Flu/RSV plus for detecting non-seasonal zoonotic influenza A viruses. While these assays are well-established for seasonal respiratory viruses, their accuracy for viruses with pandemic potential is less documented.
The key strength of this work is its focus on filling this gap. Testing these widely used platforms against non-seasonal strains demonstrates their reliability beyond seasonal diagnostics. The findings of high sensitivity/specificity are significant.
This directly supports the use of existing diagnostic tools for enhanced surveillance of potential pandemic influenza viruses.
The study makes a practical and timely contribution by broadening our understanding of these assays' capabilities for a critical public health need. The data support their role in a One Health surveillance context.
Author Response
No Comments or Suggestions were made